# Radiopharmaceutical Labelling for Lung Ventilation/Perfusion PET/CT Imaging: A Review of Production and Optimization Processes for Clinical Use

**DOI:** 10.3390/ph15050518

**Published:** 2022-04-22

**Authors:** Frédérique Blanc-Béguin, Simon Hennebicq, Philippe Robin, Raphaël Tripier, Pierre-Yves Salaün, Pierre-Yves Le Roux

**Affiliations:** 1Univ Brest, UMR-INSERM 1304, GETBO, CHRU Brest, Médecine Nucléaire, Avenue Foch, 29200 Brest, France; simon.hennebicq@chu-brest.fr (S.H.); philippe.robin@chu-brest.fr (P.R.); pierre-yves.salaun@chu-brest.fr (P.-Y.S.); pierre-yves.leroux@chu-brest.fr (P.-Y.L.R.); 2Univ Brest UMR-CNRS 6521 (CEMCA), IFR 148, Avenue Le Gorgeu, 29200 Brest, France; raphael.tripier@univ-brest.fr

**Keywords:** V/Q PET/CT, [^68^Ga]Ga-MAA, ^68^Ga-labelled carbon nanoparticles

## Abstract

Lung ventilation/perfusion (V/Q) positron emission tomography-computed tomography (PET/CT) is a promising imaging modality for regional lung function assessment. The same carrier molecules as a conventional V/Q scan (i.e., carbon nanoparticles for ventilation and macro aggregated albumin particles for perfusion) are used, but they are labeled with gallium-68 (^68^Ga) instead of technetium-99m (^99m^Tc). For both radiopharmaceuticals, various production processes have been proposed. This article discusses the challenges associated with the transition from ^99m^Tc- to ^68^Ga-labelled radiopharmaceuticals. The various production and optimization processes for both radiopharmaceuticals are reviewed and discussed for optimal clinical use.

## 1. Introduction

Lung ventilation-perfusion (V/Q) scintigraphy allows the regional lung function distribution of the two major components of gas exchanges, namely ventilation and perfusion, to be assessed [1]. Regional lung ventilation can be imaged after inhaling inert gases or radiolabelled aerosols that reach alveoli or terminal bronchioles. Regional lung perfusion can be assessed after intravenous injection of radiolabelled macroaggregated albumin (MAA) particles trapped during the first pass in the terminal pulmonary arterioles [2,3].

Pulmonary embolism (PE) diagnosis is the main clinical indication of lung V/Q scintigraphy in pulmonary embolism (PE) diagnosis. V/Q scanning was the first non-invasive test validated for PE diagnosis. The technique was then further improved with the introduction of single-photon emission computed tomography (SPECT) and, more recently, SPECT/computed tomography (CT) imaging [4]. There are many other clinical situations in which an accurate assessment of regional lung function may improve patient management besides PE diagnosis. This includes predicting post-operative pulmonary function in patients with lung cancer, radiotherapy planning to minimize the dose to the lung parenchyma with preserved function and reduce radiation-induced lung toxicities, or pre-surgical assessment of patients with severe emphysema undergoing a lung volume reduction surgery. However, although lung scintigraphy should play a central role in these clinical scenarios, its use has not been widely implemented in daily clinical practice [5]. One of the likely explanations could be the inherent technical limitations of SPECT imaging for the accurate delineation and quantification of regional ventilation and perfusion function [4].

Lung V/Q positron emission tomography (PET)/CT is a novel promising imaging modality for regional lung function assessment [6,7]. The technique has shown promising results in various clinical scenarios, including PE diagnosis [8], radiotherapy planning [9], or pre-surgical evaluation of lung cancer patients [10]. Several large prospective clinical trials are underway, such as (NCT04179539, NCT03569072, NCT04942275, and NCT05103670). The rationale is simple [5]. PET/CT uses the same carrier molecules as conventional V/Q scanning, i.e., carbon nanoparticles for ventilation imaging and MAA particles for perfusion imaging. Similar physiological processes are therefore assessed with SPECT or PET imaging. However, carrier molecules are labelled with gallium-68 (^68^Ga) instead of technetium-99m (^99m^Tc), allowing the acquisition of images with PET technology. PET has technical advantages compared with SPECT, including higher sensitivity, higher spatial and temporal resolution, superior quantitative capability and much greater access to respiratory-gated acquisition [11].

This article discusses the challenges associated with the switch from ^99m^Tc- to ^68^Ga-labelled V/Q radiopharmaceuticals. The various synthesis and optimization processes for both radiopharmaceuticals are reviewed and discussed, focusing on optimal clinical use.

## 2. Challenges of the Transition from ^99m^Tc- to ^68^Ga-Labelled Radiopharmaceuticals for Lung Imaging

As the chemical and physical properties of ^99m^Tc and ^68^Ga are different, the transition from V/Q scintigraphy to V/Q PET/CT implies some adaptation.

^99m^Tc, a metallic radionuclide, is the most widely available isotope in diagnostic nuclear medicine. It is found in oxidation states −I to VII, but the technetium (Tc) complexes for medical applications are found mostly in oxidation state V [12]. ^99m^Tc, which mainly decays (88%) with a half-life of 6.02 h by gamma emission (E_γ_ = 140 keV) to the ground state technetium-99 (^99^Tc), is obtained as ^99m^TcO_4_^−^ from a molybdenum-99 (^99^Mo)/^99m^Tc generator commercially available and compatible with the requirements of Good Manufacturing Practices (GMP). The Tc(VII) in ^99m^TcO_4_^−^ has to be reduced to a lower oxidation state to produce a ^99m^Tc-stable peptide complex or a reactive intermediate complex [13]. Tc(V) forms 5- or 6-coordinate complexes, always in the presence of multiple bond cores with heteroatoms such as oxygen (O), nitrogen (N) or sulfur (S), among which the most common in radiopharmaceuticals are the oxotechnetium and the nitridotechnetium cores [12].

The prevalent gallium (Ga) oxidation state in aqueous solution is +3, forming several gallate anions as gallium hydroxides Ga(OH)_4_^−^ at pH superior to 7. Ga(III) is a hard acid and is strongly bound to ligands featuring multiple anionic oxygen donor sites according to HSAB (hard-soft acid-base). Still, some cases have shown it to have a good affinity for thiolates [14]. Ga(III) ions can form four, five and six bonds, explaining all the possible salts or chelates. ^68^Ga decays with a half-life of 67.71 min by positron emission (88.88%) and electron capture (11.11%) to the ground state zinc-68 (^68^Zn). It is obtained from a commercially available germanium-68 (^68^Ge)/^68^Ga generator, compatible with GMP requirements.

The first challenge of the switch from ^99m^Tc- to ^68^Ga-labelled radiopharmaceuticals for lung V/Q imaging is to maintain the pharmacological properties of V and Q tracers. Both MAA and carbon nanoparticles labelled with ^99m^Tc have been largely studied. They have been shown to have a biodistribution throughout the lungs that allow an accurate assessment of regional lung perfusion and ventilation function. The principle of lung V/Q PET/CT imaging is to assess similar physiological processes than with conventional V/Q scan, but with greater technology for image acquisition.

The technique needs to be easy to implement in nuclear medicine facilities to enable routine use. The preparation should be fast, simple, GMP-compliant and safe for the operators. Furthermore, radiopharmaceutical production should use unmodified commercially available kits of carrier molecules and similar equipment and devices as much as possible as those used for conventional V/Q scans.

## 3. Lung Perfusion Imaging

### 3.1. [^99m^Tc]Tc-MAA

#### 3.1.1. Chemical Aspects of [^99m^Tc]Tc-MAA Particles

Among the various type of human serum albumin (HSA) available for radionuclide labeling, MAA is the most commonly used form in nuclear medicine facilities. The nature of the complex [^99m^Tc]Tc-MAA has not been fully elucidated. It was hypothesized that the labelling of proteins with ^99m^TcO_4_^−^ involved reduction of the anionic Tc(VII) to a cationic Tc by the tin Sn(II) contained in the commercial kit, which was then complexed with electron-donating groups [15,16,17]. Some authors have assumed that ^99m^TcO_4_^−^ reduced by the Sn(II)- albumin aggregates probably formed a (Tc = O)^3+^ complex with the aggregates [15]. More recently, high positive cooperativity was shown between ^99m^Tc and MAA, although MAA particles did not seem to have binding pockets [18,19]. Moreover, it has been shown that the speed of radiolabelling increased from HAS to albumin nanocolloids (NC) to MAA due to the greater reaction surface [18]. This result agreed with the hypothesis, which assumed that in HAS labelling kits, Sn^2+^ may be enclosed in the tertiary structure of the protein and that it may take some time for the ^99m^TcO_4_^−^ added to diffuse the site of Sn^2+^ for reduction reaction [17]. Furthermore, MAA has very complex shapes with larger surfaces than a spherical shape (as for NC) of equivalent diameter, enhancing the reactivity properties [18,19].

Whatever the exact nature of the link between ^99m^Tc and MAA, the complex [^99m^Tc]Tc-MAA demonstrates high stability as more than 90% of the radioactivity is still associated with the MAA after 24h of in vitro incubation in whole human blood at 37 °C [20].

#### 3.1.2. Technical Aspects: [^99m^Tc]Tc-MAA Preparation

[^99m^Tc]Tc-MAA particles are manually prepared by introducing a ^99m^Tc solution in a commercially available MAA kit. The ^99m^Tc is obtained from a ^99^Mo/^99m^Tc generator as sodium pertechnetate (^99m^TcO_4_^−^, Na^+^). The MAA labelling with ^99m^Tc, which occurred at pH 6, is a simple and fast (about 15 min) process, which allows the production of GMP [^99m^Tc]Tc-MAA without heating step [18].

Before intravenous administration to the patients, the [^99m^Tc]Tc-MAA suspensions are tested according to the standards mentioned by the kit supplier for clinical use. The radiochemical purity (RCP) is generally controlled using instant thin layer chromatography (iTLC), and the radioactivity distribution is assessed by filtration of the [^99m^Tc]Tc-MAA suspension through a 3-µm pore size membrane. The results are obtained by measuring filter and filtrate radioactivity. The radionuclidic purity and the pH have to be controlled as well. As MAA are large particles, [^99m^Tc], Tc-MAA must be resuspended by gentle agitation before dispensing.

#### 3.1.3. Pharmacological Aspects

In a [^99m^Tc]Tc-MAA suspension, the average particle size is 20–40 µm, and 90% have a size between 10 and 90 µm. There should be no particles larger than 150 µm [21]. [^99m^Tc]Tc-MAA particles reach the lung via the pulmonary arterial circulation. Due to the size of the alveolar capillaries (5.5 µm on average), the [^99m^Tc]Tc-MAA does not reach the alveolar capillaries but largely accumulates in the terminal pulmonary arterioles. Particles inferior to 10 µm may pass through the lungs and then phagocytose by the reticuloendothelial system [21]. According to the requirement of the MAA suppliers, the number of MAA particles injected should range from 60,000 to 700,000 to obtain uniform distribution of activity reflecting regional perfusion (for over 280 billion pulmonary capillaries and 300 million pre-capillary arterioles) [22].

Many studies have shown the suitability of the [^99m^Tc]Tc-MAA suspension to perform pulmonary perfusion scintigraphy. In rabbits, it has been shown that more than 90% of the activity was found in the lungs within a few minutes of administration and that greater than 80% of the activity remained in the lungs over the first hour of the study [20]. Malone et al. assessed the biodistribution of MAA particles in humans [23]. A total of 98% of activity was measured in the lungs immediately after injection. The removal of activity from the lungs followed an approximately bi-exponential form with the first phase in which 56% of the components had an effective half-life of 0.88 ± 0.16 h and the second phase in which 44% of the components had a half-life of 4.56 ± 0.39 h. Moreover, 3 h after injection, the [^99m^Tc]Tc-MAA uptake in the kidneys and the bladder was 3.6 ± 2.1% and 5.1 ± 4.0%, respectively [23].

### 3.2. [^68^Ga]Ga-MAA

#### 3.2.1. Chemical Aspects of [^68^Ga]Ga-MAA Particles

MAA labelling with ^68^Ga has been proposed using bifunctional chelators such as EDTA or DTPA, forming quite stable and inert chelates [24,25]. However, direct labelling was performed by most groups. Direct labelling uses a co-precipitation of ^68^Ga(III) and albumin particles [26,27]. Mathias et al. hypothesized that ^68^Ga was adsorbed to the surface of the MAA particles after hydrolysis to insoluble gallium hydroxide without excluding specific interactions of Ga(III) ion with ion pairs exposed at the particle surface [28]. As Ga is present as Ga(OH)_4_^−^ at a basic pH, ^68^Ga does not bind to MAA at a pH above 7. The MAA behavior matches with solvent-exposed glutamate and aspartate amino acids, which should be binding sites for multivalent cations with low affinity and low cation specificity [18]. Furthermore, Jain et al. found that stannous chloride (SnCl_2_) present in the MAA kits for the reduction of ^99m^Tc had a strong influence on [^68^Ga]Ga-MAA formation (radiochemical yield, mean particle diameter, serum stability), suggesting that Sn could be linked to MAA or ^68^Ga [29]. More recently, it has been assumed that MAA has multiple affinity binding sites for ^68^Ga [18]. Moreover, during ^99m^Tc and ^68^Ga competition evaluation for MAA binding sites, MAA showed no discrimination between ^99m^Tc and ^68^Ga coherently without a binding pocket [18].

#### 3.2.2. Technical Aspects: [^68^Ga]Ga-MAA Preparation

[^99m^Tc]Tc-MAA preparation is a manual and simple process involving only 2 steps: generator elution and mixing the eluate with the MAA. In contrast, because of the chemical properties of ^68^Ga, at least four steps are required to label MAA particles with ^68^Ga: ^68^Ge/^68^Ga generator elution, mixing the ^68^Ga eluate with the MAA, heating the reaction medium and the purification of the [^68^Ga]Ga-MAA. The key steps of MAA labelling with ^68^Ga are presented in Figure 1.

Table 1 summarizes the various [^68^Ga]Ga-MAA preparation processes described in the literature. The main diverging points of the preparations include (1) the choice of MAA particles, (2) the need for a ^68^Ga eluate pre-purification, (3) the labelling conditions (pH, heating temperature and time), (4) the [^68^Ga]Ga-MAA suspension purification, and (5) the automation of the process (Figure 1).

MAA

As shown in Table 1, almost all authors used commercial kits available to prepare [^99m^Tc]Tc-MAA for labelling MAA particles with ^68^Ga. The use of commercially available kits is an important consideration to facilitate the implementation of the technique in nuclear medicine facilities. However, commercially available MAA kits contain SnCl_2_ and free albumin. Consequently, many groups carried out MAA labelling with washed MAA to remove the excess of free albumin and SnCl_2_ (stannous chloride), which is usually used as a reduction component, from the kit and thus improve the labelling yields (Table 1). Ayşe et al. obtained a better final product RCP by washing the MAA particles before the labelling (RCP = 99.0 ± 0.1%) rather than not washing (RCP = 95.0 ± 0.1%) [38]. On the other hand, Mueller et al. found no significant difference in the radiolabelling yields using non-washed and pre-washed MAA (80% and 75%, respectively) [35]. In studies that used unmodified commercially available MAA kits, radiolabelling yields were consistently superior to 75.0% (Table 1) [28,35,39]. Furthermore, Jain et al. found lower radiolabelling yields using in-house synthesized MAA without SnCl_2_ than MAA with SnCl_2_ (49.9 ± 1.3% and 84.5 ± 5.3%, respectively). They found that stannous chloride present in the MAA kits used had a strong influence on the [^68^Ga]Ga-MAA formation (radiochemical yield, mean particle diameter, serum stability), suggesting that Sn could be linked to MAA or ^68^Ga [29]. Consequently, using unmodified non-washed commercially available MAA kits produced for ^99m^Tc seems to be a suitable solution for [^68^Ga]Ga-MAA labelling;

^68^Ga eluate

^68^Ga eluate obtained from currently available generators are contaminated with long-lived parent nuclide ^68^Ge and cationic metal ion impurities such as titanium (Ti)^4+^ from the column material, zinc (Zn)^2+^ from the decay of ^68^Ga or iron (Fe)^3+^. These impurities might compete with ^68^Ga in the complexation reaction [40,41].

Pre-purification of the eluate has been proposed by several groups, with various methods such as anion exchange chromatography, cationic cartridge, fractionation or eluate pre-concentration (Table 1) proposed to overcome this issue. Most groups performed an eluate pre-purification using an SCX cartridge or an equivalent pre-conditioned with hydrochloric acid (HCl) and water (Table 1).

In contrast, a few groups did not perform ^68^Ga eluate pre-purification before MAA labelling and obtained ^68^Ge impurity levels lower than 0.0001% and radiolabelling yields superior to 96.0% (Table 1) [28,29,34,39]. Based on these results and the fact that this is time-consuming, the pre-purification of the ^68^Ga eluate does not seem mandatory and could be avoided for [^68^Ga]Ga-MAA preparation.

Labelling conditions

The three key points of the MAA labelling with ^68^Ga are the pH, the heating temperature, and the reaction medium’s heating time (Figure 1).

Among the various [^68^Ga]Ga-MAA labelling processes published, the labelling pH ranges from 3 to 6.5, with the radiolabelling yield varying from 65.0 to 97.4% (decay corrected or not) (Table 1). The optimal pH range seems to be between 4 and 6.5, where ^68^Ga is a water-soluble cation [14]. Importantly, it has been shown that Ga does not bind to albumin at a pH above 7 [19]. For assessing the optimal pH for the labelling reaction, three buffers or equivalents have been used: acetate buffer, sodium acetate solution and HEPES buffer (Table 1). HEPES and acetate buffers are biocompatible with no toxicity issue [40]. They have demonstrated better characteristics to stabilize and prevent ^68^Ga(III) precipitation and colloid formation [40]. Nevertheless, in contrast to sodium acetate, HEPES is not approved for human use, and thus, purification and additional quality control analyses are required, resulting in further time and resource consumption [40].

The labelling temperatures reported in the literature ranged from 40 °C to 115 °C (Table 1). Whatever the labelling conditions, the radiolabelling yields were superior to 75.0% in all but one study (Table 1). Several studies have tested various labelling conditions and obtained increasing radiolabelling yield by increasing the heating temperature (from 25 to 100 °C, pH: 4–6) [29,38]. On the other hand, low heating temperature (40–70 °C) has been described to preserve MAA structure and size because high temperatures may induce the rupture of bigger macroaggregates [34,36,39]. Accordingly, the heating temperature range seems quite large, as long as MAA integrity is maintained.

Finally, the labelling heating time ranged from 5 to 20 min (Table 1). Some studies have compared various heating times (from 3 to 50 min) and found higher labelling yields with heating times between 7 and 20 min [31,38,39]. Due to the short half-life of ^68^Ga and for practical considerations, the heating step should be as short as possible.

[^68^Ga]Ga-MAA purification

Many authors have shown that the final product RCP was improved by performing a purification step at the end of the labelling (Table 2). Various processes were used to purify [^68^Ga]Ga-MAA suspension. The most commonly used process was to wash the labelled MAA with saline and centrifuge to isolate [^68^Ga]Ga-MAA particles. Another process was to increase the reaction mixture to 10 mL with sterile water at the end of the heating step and to pass the suspension through a Sep-Pak C18 cartridge. Then, the cartridge was washed two times with sterile water. The RCP of the obtained final product was superior to 97% (Table 2).

While purification processes have only been manual to date, an innovative automated procedure was recently proposed for the purification step using a low protein-binding filter. At the end of the heating step, the reaction medium was passed through the filter from the bottom. Then, [^68^Ga]Ga-MAA was removed from the filter and transferred into the final vial by passing 10 mL of saline from the top to the bottom of the filter (see Figure 2) [39]. The RCP of the obtained [^68^Ga]Ga-MAA suspension was 99.0 ± 0.6%.

Manual or Automated Process

Most of the [^68^Ga]Ga-MAA preparation processes described in the literature were manual and were carried out in 20 to 40 min (See Table 1) [28,31,32]. Mueller et al. showed that an automated process reduced the preparation time from 20 to 14 min [35]. A few automated processes were described in the literature to perform the synthesis from the elution to the end of the heating steps (see Table 1). More recently, a fully automated process has been proposed, including both MAA labelling and [^68^Ga]Ga-MAA purification, which was carried out in 15 min (see Table 1 and Table 2) [39]. To the best of our knowledge, this is the only fully automated process to date.

#### 3.2.3. Pharmacological Aspects

An important challenge of the switch from ^99m^ Tc- to ^68^Ga-labelled MAA is maintaining the pharmacological properties of particles to ensure similar biodistribution throughout the terminal pulmonary arterioles. Accordingly, the key parameter is the particle size, which should range between 10.0 and 90.0 µm, with no particles size superior to 150.0 µm. On the other hand, particles should not be inferior to 10.0 µm because the target organs would be the reticuloendothelial system and the bones instead of the lungs [22]. Most of the literature data reported a mean diameter ranging from 10 to 90 µm (15.0–75.0 µm for Blanc-Béguin et al., 52.9 ± 15.2 for Jain et al. and 43.0–51.0 for Canziani et al.) (Table 3) [18,29,39]. Furthermore, all published data on [^68^Ga]Ga-MAA described a preserved structure of radiolabelled particles, whatever the labelling conditions (see Appendix A).

Hence, [^99m^Tc]Tc-MAA and [^68^Ga]Ga-MAA particles have similar sizes and structures. The number of [^68^Ga]Ga-MAA particles injected should range from 60,000 to 700,000, no differently from [^99m^Tc]Tc-MAA, to obtain uniform distribution of activity reflecting regional perfusion.

All studies that assessed the biodistribution of labelled MAA particles in the animals described an almost complete retainment of activity in the lungs from 5 min to 30, 45, 60 min, or 4 h after the injection of [^68^Ga]Ga-MAA particles, and very low activity in other organs, especially in the liver [22,29,33,34]. Indeed, experiments performed with rats or mice have shown that more than 80% of the injected activity was located in the lungs from 15 min to 4 h post-injection [22,29,34]. In female wild-type rats, the peak of activity occurred 1 min and 35 min post-injection in the kidneys and bladder, respectively, whereas it was from 10 to 20 min in the lungs [22]. In Sprague-Dawley rats, less than 2% of the injected dose per organ (ID/o) activity was measured in other organs from 2 to 4 h post-[^68^Ga]Ga-MAA administration in the tail vein [34]. It is noteworthy that, as compared with the [^99m^Tc]Tc-MAA, [^68^Ga]Ga-MAA exhibited better in vivo stability after intravenous injection in Sprague-Dawley rats [34]. Indeed, the percentage of decay-corrected ID/o (DC-ID/o) of [^68^Ga]Ga-MAA located in the lung did not change over the study period, i.e., the 4 h following the injection (98.6 ± 0.7 at 2 h and 98.6 ± 0.1 at 4h), whereas the % DC-ID/o of [^99m^Tc]Tc-MAA located in the lung decreased from 86.6 ± 0.7 at 2 h to 79.2 ± 1.5 at 4 h [34]. After injection in the rat tail vein, the main activity was extracted by the kidneys to the bladder, and the free ^68^Ga remained in the blood after two and four hours (84.9 ± 4.5% and 63.1 ± 3.9% of DC-ID/o, respectively), presumably as ^68^Ga native transferrin complex [22,34].

^68^Ga(III) has a high binding affinity to the blood serum protein transferrin (log K1 = 20.3). The main requirement for [^68^Ga]Ga-MAA stability is thermodynamic stability towards hydrolysis and formation of Ga(OH)_3_ [42]. As shown in Table 3, whatever the labelling conditions, the obtained [^68^Ga]Ga-MAA suspension had at least a 45 min in vitro stability in animal serum or plasma, which is largely sufficient given that images are acquired immediately after injection and that the acquisition time is approximately 5 min [33].

To the best of our knowledge, no data were published on the biodistribution of [^68^Ga]Ga-MAA in humans. However, Ament et al., who performed an exploratory study on five patients with clinical suspicion of PE who underwent V/Q PET/CT, have observed that perfusion imaging was homogeneous in most cases [33]. No significant retention and no visual uptake of [^68^Ga]Ga-MAA particles in the liver were detectable [33].

However, many clinical studies reported consistent activity distribution on PET imaging in various pulmonary conditions after injection of [^68^Ga]Ga-MAA [1,5,8,33,35,43].

## 4. Lung Ventilation Imaging

### 4.1. Aerosolized ^99m^Tc-Labelled Carbon Nanoparticles (Technegas)

#### 4.1.1. Physical and Chemical Aspects

^99m^Tc-labelled carbon nanoparticles consist of primary hexagonally structured carbon nanoparticles, which can agglomerate into larger secondary aggregates. Primary nanoparticles are structured with graphite planes oriented parallel to the technetium surface to form nanoparticles with a thickness of about 5 nm [44]. Few data are available about the link between ^99m^Tc and carbon nanoparticles. It was hypothesized that Tc^7+^ obtained from a ^99^Mo/^99m^Tc generator was reduced at the crucible interface, resulting in native metal Tc which co-condensates with carbon species once in the vapor phase [44].

Some authors have hypothesized that ^99m^Tc-labelled carbon nanoparticles consisted of ^99m^Tc atoms trapped by a structure similar to a fullerene cage [45,46]. Indeed, Mackey et al. have demonstrated the presence of fullerenes during the generation of the aerosolized ^99m^Tc-labelled carbon nanoparticles available to form metallofullerenes with the ^99m^Tc atom attached either exohedrally or endohedrally to the fullerene molecules [46]. However, this hypothesis was controversial especially because of the hexagonal platelet structure of the labelled carbon nanoparticles [44].

#### 4.1.2. Technical Aspects

The ^99m^Tc-labelled carbon nanoparticle production is a simple process that requires relatively little material: a ^99^Mo/^99m^Tc generator, a Technegas generator (Cyclomedica Pty Ltd., Kingsgrove, Australia) and a pure argon bottle. There are three main stages in ^99m^Tc-labelled carbon nanoparticle production: the loading of the crucible, the simmer stage and the burning stage.

Crucible loading

Using a syringe with a needle, 0.14 mL to 0.30 mL (140–925 MBq) of ^99m^Tc eluate is introduced in a graphite crucible previously humidified with 99% ethanol to increase its wettability and placed between the generator electrodes [6,47,48,49,50,51]. As the volume of the crucible is limited to 0.14 mL or 0.30 mL according to the supplier, it is possible to perform several crucible loadings to introduce all the ^99m^Tc eluate needed;

Simmer stage

The simmer stage, performed immediately after the crucible loading, reduces ^99m^Tc^7+^ to metallic ^99m^Tc under a pure argon atmosphere [44]. The use of pure argon is a determining parameter for the structure and the physical properties of the ^99m^Tc-labelled carbon nanoparticles [45,48,52,53,54]. During the simmer stage, the graphite crucible is heated for 6 min at 70 °C. However, it has been shown that increasing the number of simmers increases the median size of the ^99m^Tc-labelled carbon nanoparticles [45];

Burning stage

The simmer cycle is followed by the crucible heating to 2550 °C ± 50 °C for 15 s. Metallic Tc and carbon species are vaporized and co-condensed during this burning stage to obtain aerosolized ^99m^Tc-labelled carbon nanoparticles [44]. At the end of this stage, the switching off of the Technegas generator fills the 6 L chamber with aerosolized ^99m^Tc-labelled carbon nanoparticles ready for use via inhalation by the patient.

All process parameters (heating temperature and time) used for clinical production are fixed. The machine allows a 10 min window in which the ^99m^Tc-labelled carbon nanoparticles may be administered to the patient. However, the longer the administration delay is, the higher the median size of the particles is [45,51].

#### 4.1.3. Pharmacological Aspects

The size of primary carbon nanoparticles ranges from 5 to 60 nm, while the size of the aggregates is approximately 100–200 nm (See Table 4). Hence, aerosolized ^99m^Tc-labelled carbon nanoparticles are considered an ultrafine aerosol with ventilation properties similar to radioactive gasses, such as krypton-81m (^81m^Kr) and xenon-133 (^133^Xe) [21,47,48,50,51,55,56,57]. Many authors agree on the mainly alveolar deposition of the ^99m^Tc-labelled carbon nanoparticles and the stability of the nanoparticles in the lungs over time [21,45,49,57,58].

### 4.2. Aerosolized ^68^Ga-Labelled Carbon Nanoparticles

#### 4.2.1. Physical and Chemical Aspects

The physical properties of aerosolized particles are important parameters in determining their penetration, deposition, and retention in the respiratory tract. The physical properties of ^68^Ga-labelled carbon nanoparticles, prepared using a Technegas generator in the usual clinical way, were recently assessed [60]. ^68^Ga-labelled carbon nanoparticles demonstrated similar properties as ^99m^Tc-labelled carbon nanoparticles, with primary hexagonally shaped and layered structured particles [60]. Although the chemical process of labelling carbon nanoparticles with ^68^Ga and the exact chelation structure of ^68^Ga in carbon nanoparticles are unknown, the physical properties of ^68^Ga-labelled carbon nanoparticles suggest a method of labelling similar to labeling with ^99m^Tc.

#### 4.2.2. Technical Aspects

In contrast with MAA labelling, the process for ^99m^Tc-labelled carbon nanoparticle preparation is very similar across studies in the literature. ^68^Ga-labelled carbon nanoparticles are produced using an unmodified Technegas generator and following the same stages as for the preparation of ^99m^Tc-labelled carbon nanoparticles: the crucible loading with an eluate volume range from 0.14 mL to 0.30 mL, the simmer stage and the burning stage with similar heating time and temperature [6,7,33,60,61,62]. The only difference is the nature of the eluate, which is gallium-68 chloride (^68^GaCl_3_) instead of ^99m^TcO_4_^−^, Na^+^. Few authors performed an eluate pre-concentration using an anion exchange cartridge or by fractionating to purify and reduce the volume of the eluate [33,62]. However, ^68^Ga-labelled carbon nanoparticles obtained using an unmodified eluate from the ^68^Ge/^68^Ga generator demonstrated similar physical properties as ^99m^Tc-labelled carbon nanoparticle properties and were suitable for pulmonary ventilation PET/CT [6,7,60,61].

#### 4.2.3. Pharmacological Aspects

From the pharmacological point of view, an important parameter of the switch from lung ventilation SPECT to PET/CT is to maintain the physical properties of aerosolized carbon nanoparticles to ensure similar alveolar deposition and stability in the lungs.

The size is a key factor in determining the degree of aerosol particle penetration in the human pulmonary tract [55]. To the best of our knowledge, only one recently published work has studied the physical properties of ^68^Ga-labelled carbon nanoparticles, and few pharmacological data are available in the literature. However, it was reported that using an unmodified Technegas generator, the mean diameter of primary ^68^Ga-labelled carbon nanoparticles was in the same range as primary ^99m^Tc-labelled carbon nanoparticles (22.4 ± 10.0 nm and 20.9 ± 7.2 nm, respectively) with similar agglomeration into larger secondary aggregates measuring several hundreds of nm [60].

These results suggested similar lung distribution of ^99m^Tc- and ^68^Ga-labelled carbon nanoparticles, as confirmed by a study on healthy piglets [62]. Later, V/Q PET performed in Sprague-Dawley rats reported complete incorporation of ^68^Ga-labelled carbon nanoparticles in the lungs without extrapulmonary activity (urinary bladder, abdomen, blood pool) [33]. Moreover, animal studies performed with aerosolized ^68^Ga-labelled carbon nanoparticles have demonstrated greater differences between poorly and well-ventilated regions, suggesting higher resolution than ^99m^Tc-labelled carbon nanoparticles [62].

Moreover, in healthy human volunteers, the activity distribution in the lungs after inhalation of ^68^Ga-labelled carbon nanoparticles was intense, without bronchial deposit 15 min after the inhalation. Furthermore, the activity (decay corrected) over the lung was constant at 3.5 h without elimination via blood, urine (only trace radioactivity in urine bladder was observed) or feces suggesting the stability of the deposition of ^68^Ga-labelled carbon nanoparticles over this time [7]. Another exploratory study performed on five patients with clinical suspicion of PE observed homogeneous ventilation imaging in two cases and inhomogeneous accumulation with central deposition of labelled carbon nanoparticles in three cases [33].

Finally, many clinical studies reported consistent activity distribution in the lungs on PET imaging in various pulmonary conditions after inhalation of ^68^Ga-labelled carbon nanoparticles [5,6,8,33].

## 5. Practical Considerations for an Optimal Clinical Use

Lung V/Q PET/CT is a promising imaging modality for regional lung function assessment. Indeed, PET imaging has great technical advantages over SPECT imaging (higher sensitivity, spatial and temporal resolution, superior quantitative capability, easier to perform respiratory-gated acquisition). PET may also be a useful alternative to SPECT imaging in a ^99m^Tc shortage. The success of the switch from conventional scintigraphy to PET imaging, and therefore from ^99m^Tc- to ^68^Ga-labelled radiopharmaceuticals, relies on two main factors: preserving the pharmacological properties of the labelled MAA and carbon nanoparticles, whose biodistribution is well known; and facilitating the implementation in nuclear medicine departments. In that respect, several studies have been conducted on the production of both perfusion and ventilation ^68^Ga-labelled radiopharmaceuticals, which have led to simplification, optimization and, more recently, automation of the processes.

For lung perfusion PET/CT imaging, various processes have been used for [^68^Ga]Ga-MAA labelling, with different options in the key steps of the preparation, including the choice of MAA particles, the need for ^68^Ga eluate pre-purification, the labelling conditions or the [^68^Ga]Ga-MAA suspension purification. However, simpler processes appear to be suitable for optimal clinical use. This includes using a non-modified commercially available MAA kit, with no need for a ^68^Ga eluate pre-purification, use of an easy to use buffer such as sodium acetate solution, and a short reaction medium heating time (5 min). Automated processes have been developed to facilitate processing time and reduce the radiation dose to the operator. Thus, a simple and fast (15 min) automated GMP compliant [^68^Ga]Ga-MAA synthesis process was proposed, using a non-modified MAA commercial kit, a ^68^Ga eluate without pre-purification and including an innovative process for [^68^Ga]Ga-MAA purification, which maintains the pharmacological properties of the tracer and provided labelling yields >95% [39]. Moreover, whatever the labelling conditions, the obtained [^68^Ga]Ga-MAA suspension was described to be stable in 0.9% sodium chloride for at least one hour [35,39]. Given the radioactive concentration of [^68^Ga], Ga-MAA obtained at the end of the synthesis (i.e., from 300MBq/10 mL to 900 MBq/10 mL according to the age of the ^68^Ge/^68^Ga generator) and the dose injected (i.e., around 50 MBq), up to 6 perfusion PET/CT scans can be performed with one synthesis [5,6,39,63].

For lung ventilation PET/CT imaging, preparing and administering aerosolized ^68^Ga-labelled carbon nanoparticles is very straightforward. The process is very similar to the production of ^99m^Tc-labelled carbon nanoparticles and, therefore, fairly easy to implement in nuclear medicine facilities. Indeed, adding a ^68^Ga eluate instead of ^99m^Tc eluate in the carbon crucible of an unmodified commercially available Technegas™ generator provides carbon nanoparticles with similar physical properties. Furthermore, recently, an automated process included a step to fractionate the ^68^Ga eluate into two samples, one for [^68^Ga]Ga-MAA labelling and the other for aerosolized ^68^Ga-labelled carbon nanoparticle production, which has been developed [39].

Besides radiopharmaceutical production, many factors may facilitate the implementation of V/P PET/CT imaging in nuclear medicine facilities. ^68^Ge/^68^Ga generators are increasingly available in the nuclear medicine departments due to ^68^Ga tracers for neuroendocrine tumors and prostate cancer imaging. PET/CT cameras are also increasingly accessible due to the development of digital PET/CT cameras and might be total-body PET/CT in the future. Most nuclear medicine facilities already have the necessary equipment to carry-out V/P PET/CT imaging, including carbon nanoparticle generators and MAA kits. Automating the MAA labelling is now possible; commercial development of ready-to-use sets for automated synthesis radiolabelling of ^68^Ga-MAA would be of interest.

In conclusion, recent data support the ease of using well-established carrier molecules and ^68^Ga to enable the switch from SPECT to PET imaging for regional lung function. The technology may be easily implemented in most nuclear medicine facilities and open perspectives for the improved management of patients with lung disease.

## Figures and Tables

**Figure 1 pharmaceuticals-15-00518-f001:**
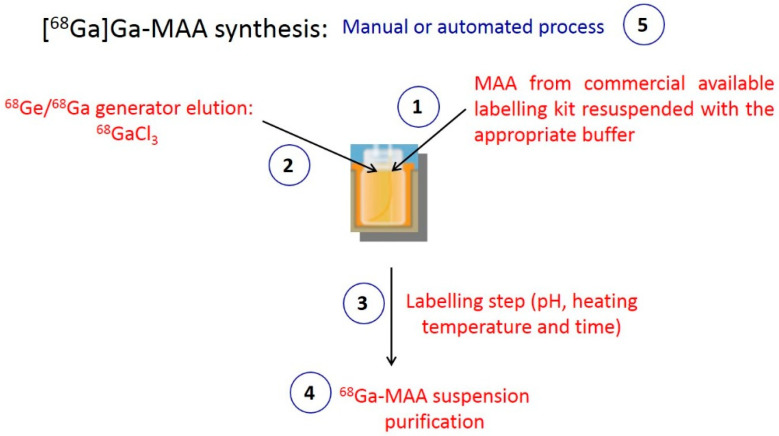
Key points of MAA labelling with ^68^Ga.

**Figure 2 pharmaceuticals-15-00518-f002:**
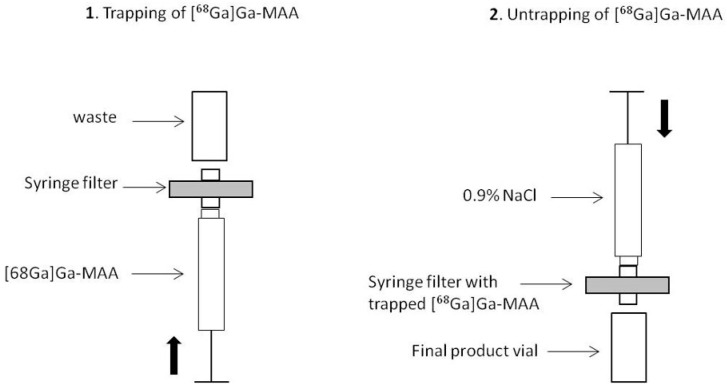
Schematic representation of the [^68^Ga]Ga-MAA purification stage based on the process by Blanc-Béguin et al. Used with permission from Blanc-Béguin et al. [39].

**Table 1 pharmaceuticals-15-00518-t001:** [^68^Ga]Ga-MAA labelling conditions of various methods described in the literature.

	Albumin Particles Labelled		^68^Ga Eluate Pre-Purification		Labelling Conditions				Radiolabelling Yield (%)	Process Duration (min)	Manual/Automated Process
	Nature and Origin	Washed/Non-Washed	Yes/No	Method Used	pH	Buffer	Heating Time (min)	T °C			
Hnatowich et al. [26]	HSA microspheres. Commercial kit.	Non-washed	Yes	Anion exchange chromatography	2.6–3		15	40–60	65.0	40	Manual
Hayes et al. [30]	HSA microspheres. Commercial kit.	Non-washed	Yes	Anion exchange chromatography	4.8	Sodium acetate solution	10	37	97.4 ± 1.0		Manual
Maziere et al. [31]	HSA microspheres. Commercial kit.	Non-washed	Yes	Pre-concentration	4.7		20	85	93.2 ± 2.5	40	Manual
Even and Green [24]	MAA. Commercial kit.	Washed	Yes	Pre-concentration	4.7	Sodium acetate buffer	15	74 ± 1	77.0–97.0		Manual
Mathias et al. [28]	MAA. Commercial kit.	Washed			5–6	Sodium acetate solution	15	75	81.6 ± 5.3	25	Manual
Maus et al. [32]	MAA. Commercial kit.	Washed	Yes	Fractionation	4	HEPES	20	75	85.0 ± 2.0	30	Manual
Hofman et al. [6]	MAA. Commercial kit.	Washed	Yes	SCX cartridge	6.5	Sodium acetate solution	5	70	≥90.0		Manual
Ament et al. [33]	MAA. Commercial kit.	Washed	Yes	Fractionation	4	HEPES	20	75	85.0 ± 2.0		Manual
Amor-Coarasa et al. [34]	MAA. Commercial kit.	Washed	No		4.7	Sodium acetate solution	15	75	78.3 ± 3.1		Manual
			Yes	Combination of chromatographic exchange resins (cationic then anionic)	4.7	Sodium acetate solution	15	75	97.6 ± 1.5		Manual
Jain et al. [29]	MAA. Homemade with and without SnCl_2_		No		6		15	75	77.6		Manual
Mueller et al. [35]	MAA. Commercial kit.	Washed and non-washed	Yes	SCX cartridge	4.5	Sodium acetate buffer	10	115	Not mentioned (non-washed) 75.0 (washed)	20 (manual) 14 (automated)	Manual and automated
Shannehsazzadeh et al. [22]	MAA. Commercial kit.	Washed	Yes	Fractionation	4	HEPES	8	75	90.0 -95.0		Manual
Persico et al. [36]	MAA. Commercial kit.	Washed	Yes	SCX cartridge	6–6.5	Sodium acetate buffer	15	40	97.0		Manual and automated
Gultekin et al. [37]	MAA. Commercial kit.	Washed	Yes	PSH^+^ cartridge	4–5	HEPES	7	90	80.0		Manual
Ayșe et al. [38]	MAA. Commercial kit.	Washed	Yes	PSH^+^ cartridge	4–5	HEPES	7	90	85.0 ± 3.0	16	Automated
Blanc-Béguin et al. [39]	MAA. Commercial kit.	Non washed	No		4.3	Sodium acetate solution	5	60	96.0 ± 1.7	15	Automated

**Table 2 pharmaceuticals-15-00518-t002:** Process of [^68^Ga]Ga-MAA purification and radiochemical purity according to the various authors.

	[^68^Ga]Ga-MAA Purification Conditions		Radiochemical Purity (%)
	Process of Purification	Manual/Automated	
Maziere et al. [31]	Centrifugation	Manual	99.9
Even and Green [24]	Centrifugation	Manual	89.0 ± 5.0–98.4 ± 0.3
Mathias et al. [28]	Centrifugation	Manual	99.8 ± 0.1
Maus et al. [32]	Sep-Pak C_18_ cartridge	Manual	>97.0
Ament et al. [33]	Centrifugation	Manual	>97.0
Amor-Coarasa et al. [34]	Centrifugation	Manual	>95.0
Jain et al. [29]	Centrifugation	Manual	98.0 ± 0.8
Mueller et al. [35]	No purification		>95.0
Shannehsazzadeh et al. [22]	Centrifugation	Manual	100.0
Persico et al. [36]	No purification		97.0
Gultekin et al. [37]	Centrifugation	Manual	99.0
Ayșe et al. [38]	No purification		99.0
Blanc-Béguin et al. [39]	Filtration	Automated	99.0 ± 0.6

**Table 3 pharmaceuticals-15-00518-t003:** Summary of important factors of the switch from ^99m^Tc to ^68^Ga.

		[^99m^Tc]Tc-MAA	[^68^Ga]Ga-MAA
**Labelling conditions**			
	pH	6	4–6.5
	Heating temperature (°C)	Room temperature	40–115 °C
	Heating time (min)	0	5–20 min
	Total labelling time	20 min	15–40 min
**Size**			
	µm	10.0–90.0	15.0–75.0
**Labelled MAA suspension stability**			
	hours	8	3
**Labelled MAA in vitro serum stability**			
	hours	24	1
**Biodistribution**		In humans	In animals
Lungs uptake	%	98.0 [23]	
	time	Immediately after injection	
	%	86.6 ± 0.7 [34]	98.6 ± 0.7 [34]
	time	2 h post-injection	
Kidney uptake	%	3.6 ± 2.1 [23]	1.6 ± 0.4 (right kidney), 1.4 ± 0.2 (left kidney) [34]
	time	3 h after injection	4 h after injection
Bladder uptake	%	5.1 ± 4.0	14 ± 1.7
	time	3 h after injection	4 h after injection [34]
Stomach uptake	%	3.5 ± 2.5	
	time	4.4 after injection	

**Table 4 pharmaceuticals-15-00518-t004:** ^99m^Tc and ^68^Ga-labelled carbon nanoparticle size, shape and structure according to the literature.

		Labelled Carbon Primary Nanoparticle Size (nm)	Labelled Carbon Secondary Aggregate Size (nm)	Count Median Diameter (nm)	Labelled Carbon Nanoparticle Thickness (nm)	Shape, Structure and Properties	Physical Properties
** ^99m^ ** **Tc labelling**	Burch et al. [47]	≤5.0					
	Strong et al. [58]			140.0 ± 1.5			
	Isawa et al. [49]	≤200.0					
	Lemb et al. [59]	12.5 ± 1.65 (7–23)	118.0 (60–160)			Primary hexagonally structured graphite particles	Hydrophobic properties. Inert properties
	Mackey et al. [46]					Fullerenes	
	Lloyd et al. [45]		100.0–300.0	158.0 ± 1.5			
	Senden et al. [44]	30.0–60.0			5.0	Thin hexagonal platelets with graphite planes oriented parallel to the Tc surface	Biological inertness
	Möller et al. [57]	10.0	100.0–200.0				Hygroscopic properties
	Pourchez et al. [51]	40.0 ± 2.9				Hexagonal platelets of metallic technetium closely encapsulated with a thin layer of graphitic carbon	Hygroscopic properties
	Blanc-Béguin et al. [60]	20.9 ± 7.2				Thin hexagonal platelets with graphite planes oriented parallel to the Tc surface	
** ^68^ ** **Ga labelling**	Blanc-Béguin et al. [60]	22.4 ± 10	Several hundreds			Hexagonal shape Layered structure	

## Data Availability

Not applicable.

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
