# Peer review of "Radiopharmaceutical Labelling for Lung Ventilation/Perfusion PET/CT Imaging: A Review of Production and Optimization Processes for Clinical Use"

_pharmaceuticals, 2022, doi:10.3390/ph15050518_

Round 1
Reviewer 1 Report
This manuscript describe a review of radiopharmaceuticals labelling with 68Ga instead of 99mTc for lung ventilation/perfusion PET/CT imaging. This review manuscript is well written in terms of the production methods of 68Ga labelled agents in detail. Therefore, it should be acceptable for publishing in pharmaceuticals after the minor revision as below.
- In page 2, line 87, the mass number of "68Ge/68Ga" should be superscript. I also found the same things in the following pages, so there should be also corrected.
- In page 8, ling 206-208, the RCP value of 68Ga-MAA derived from reference 39 seemed to be different to that in table 1. Which values were true?
- In table 1, some reports (i.e. ref 28 and 35) compared the two method (washing or without washing of 68Ga-MAA) but only one RCY value was described. The both values should be written.
- In page 10, line 343, about the distribution of 68Ga-MAA in rat model, the unit of the radioactivity in blood was described as "%", but was it "%ID/g"?
- In table 4, it included only the information of 99mTc-labelled carbon nanoparticles. The main theme of this review manuscript seems to be 68Ga labelled agents; therefore the information of 68Ga labelled carbon nanoparticles should be added and compared with those of 99mTc-labelled agents, or if it is quite difficult to be added, this table should be in the supporting information sections.
Reviewer 2 Report
The authors present the labeling method, purification, pharmacological aspects, Practical considerations of 68Ga-labelled MAA and 68Ga-labelled carbon nanoparticles. The contents of this paper are good, and all parts of the paper are well written. However, more information on pharmacological aspects is needed to help understand readers who have an interest in the related field. Currently, I recommend publication after minor revisions. The addition of the details will make the manuscript much more suitable for publication.
- Provide a figure for a chelation structure of 99mTc and 68Ga in MAA and carbon nanoparticles.
- The paper's authors argue that pharmacological aspects of 68Ga-labelled MAA and 68Ga-labelled carbon nanoparticles are important. However, information on pharmacological aspects is too short for both agents. Please provide more details about that as a review paper.
- Lines 18-19: “Ga-68, Tc-99m” change to 68Ga and 99mTc
- Lines 117, 350: “in vitro” change to italic
- Lines 48: “Computed Tomography” is already mentioned above.
- Lines 340: The explanation or reason is needed for “better in vivo stability” of Ga-MAA
- Lines 451: superscript (68Ge/68Ga)
